# Prognostic Value of Inflammatory and Nutritional Biomarkers of Immune Checkpoint Inhibitor Treatment for Recurrent or Metastatic Squamous Cell Carcinoma of the Head and Neck

**DOI:** 10.3390/cancers15072021

**Published:** 2023-03-28

**Authors:** Akihiro Sakai, Hiroaki Iijima, Koji Ebisumoto, Mayu Yamauchi, Takanobu Teramura, Aritomo Yamazaki, Takane Watanabe, Toshihide Inagi, Daisuke Maki, Kenji Okami

**Affiliations:** Department of Otolaryngology, Head and Neck Surgery, School of Medicine, Tokai University, Isehara 259-1193, Japan

**Keywords:** inflammation, nutrition, biomarker, immune checkpoint inhibitor, head and neck cancer

## Abstract

**Simple Summary:**

In recent years, various biomarkers have been developed to assist in the selection of anticancer agents. Inflammatory and nutritional biomarkers have been reported as useful for predicting prognosis after treatment with immune checkpoint inhibitor (ICI) treatment in head and neck cancers. Still, their prognostic value in recurrent or metastatic squamous cell carcinoma of the head and neck (HNSCC) has not been thoroughly investigated. Therefore, we evaluated the prognostic value of inflammatory and nutritional biomarkers of ICI treatment for RMHNSCC. We demonstrated that the lymphocyte-to-monocyte ratio (LMR) was the most important biomarker. This study suggests that LMR may be the most useful biomarker for predicting the prognosis of ICI treatment for RMHNSCC.

**Abstract:**

This study aimed to determine the prognostic value of inflammatory and nutritional biomarkers of immune checkpoint inhibitor (ICI) therapy for recurrent or metastatic squamous cell carcinoma of the head and neck (RMHNSCC) and to identify the most useful factor for prognosis assessment. We retrospectively reviewed the medical records of patients with RMHNSCC who received ICI therapy. The response rate for ICI therapy and the relationship between inflammatory and nutritional biomarkers and overall survival were examined. The included biomarkers did not correlate with an objective response rate but were associated with a disease control rate. Univariate analysis showed significant correlations between the serum albumin level, C-reactive protein level, platelet to lymphocyte ratio, neutrophil to lymphocyte ratio, lymphocyte to monocyte ratio (LMR), systemic immune-inflammation index, and controlling the nutritional status score and overall survival; multivariate analysis showed that LMR was significantly correlated with overall survival. LMR was the most important biomarker according to the machine learning model. This study suggests that LMR may be the most useful biomarker for predicting the prognosis of ICI treatment for RMHNSCC.

## 1. Introduction

Head and neck squamous cell carcinomas constitute the sixth most common cancer type in the world [1]. Despite improvements in treatment, recurrence, and metastasis are common and contribute to poor prognosis. Therefore, improving the prognosis of recurrent or metastatic squamous cell carcinoma of head and neck (RMHNSCC) patients is crucial for the treatment of RMHNSCC. Recently, approved immune checkpoint inhibitors (ICIs) have significantly improved the prognosis of patients with recurrent or metastatic squamous cell carcinoma of the head and neck (RMHNSCC) compared to conventional chemotherapy [2,3]. ICIs have been widely used as a first-line treatment for RMHNSCC because they are associated with fewer adverse events [4,5] and better patient quality of life than chemotherapy. NCCN Guidelines for Head and Neck Cancers V.1.2023 recommend two ICIs, nivolumab, and pembrolizumab, as the preferred regimens. The CheckMate 141 study [2] was a clinical trial that evaluated the effectiveness of nivolumab for RMHNSCC. The trial compared nivolumab to chemotherapy and found that patients who received nivolumab had longer overall survival rates and fewer adverse events than chemotherapy. Additionally, the Keynote-048 trial [3] reported the effectiveness of pembrolizumab as a first-line treatment for RMHNSCC. Pembrolizumab alone and pembrolizumab plus chemotherapy demonstrated superior overall survival (OS) and progression-free survival (PFS) compared to chemotherapy, leading to the approval of pembrolizumab as a first-line treatment for RMHNSCC. However, the response rates in the Checkmate 141 study [2] and the Keynote 048 study [3] were 13.3% and 16.9%, respectively, and not many patients benefit from ICIs. Therefore, estimating the effects of ICIs and prognosis after ICI treatment is crucial in determining the optimal treatment strategy.

In recent years, various biomarkers have been developed to assist in the selection of anticancer agents [6,7], including PD-L1 expression [8,9], tumor mutation burden [10], interferon-γ signature [11], and the tumor microenvironment [12]. However, most of these biomarkers are research-based and unsuitable for clinical application. By contrast, several inflammatory and nutritional biomarkers have been reported to be associated with prognosis [13,14,15,16,17]. In particular, the neutrophil-to-lymphocyte ratio (NLR) is a well-known biomarker that has been reported to be an independent prognostic factor in head and neck cancer [18]. In addition, many inflammatory and nutritional biomarkers that can be easily measured from blood tests, such as the platelet-to-lymphocyte ratio (PLR) [19,20], lymphocyte-to-monocyte ratio (LMR) [21], systemic immune-inflammation index (SII) [22,23], CRP-to-Alb ratio (CAR) [24,25], the controlling nutritional status (CONUT) score [14,26], prognostic nutrition index (PNI) [27,28], prognostic index (PI) [29], and the Glasgow Prognostic Score (GPS) [13], have been reported to be useful for predicting prognosis after treatment in head and neck and other cancers [19,20,24,25,30]. However, it is unclear which biomarkers best predict prognosis after ICI treatment, and their prognostic value in RMHNSCC has not been thoroughly investigated.

Therefore, this study aimed to determine the prognostic value of inflammatory and nutritional biomarkers, the levels of which can be measured by blood analysis in the treatment of RMHNSCC with ICIs, and to identify the most useful factor for prognosis assessment.

## 2. Materials and Methods

### 2.1. Patient and Data Collection

We retrospectively reviewed the medical records of patients with RMHNSCC who underwent ICI therapy at Tokai University Hospital in Kanagawa, Japan, from April 2017 to June 2022. Patients were included if they had an Eastern Cooperative Oncology Group (ECOG) performance status (PS) of 0–2, had received at least one cycle of immunotherapy, had adequate organ function, and could undergo imaging analysis or be clinically evaluated after ICI therapy. Nivolumab was administered to patients every two weeks at 3 mg/kg of body weight or 240 mg/body doses. Pembrolizumab was administered at a dose of 200 mg every three weeks. The ICI was selected at our multidisciplinary head and neck cancer conference based on each patient’s condition. In principle, nivolumab was indicated for the treatment of platinum-refractory RMHNSCC, and pembrolizumab was indicated for the first-line treatment of RMHNSCC or platinum-sensitive RMHNSCC. ICIs were continued until disease progression or unacceptable toxicity was observed, and patients were followed up until death or the cutoff date (31 May 2022).

The clinical response to treatment was assessed every 4–12 weeks using computed tomography. Tumor response was evaluated using the Response Evaluation Criteria in Solid Tumors (RECIST) version 1.1. The objective response rate (ORR) was defined as the percentage of patients who achieved a complete response (CR) or a partial response (PR) as the best response. The disease control rate (DCR) was defined as the percentage of patients with CR, PR, or stable disease (SD) as the best response. OS was defined as the time from the start of treatment to the date of death, regardless of the cause or cutoff. Progression-free survival (PFS) was defined as the time from the start of treatment to the cutoff date when disease progression or death for any reason was no longer observed. The duration of response was defined as the time from the initial response (CR, PR, or SD) to the start of disease progression (PD).

Adverse events were recorded using the National Cancer Institute Common Terminology Criteria for Adverse Events version 5.0.

The Institutional Review Board of Tokai University Hospital (22R200) approved this study, which was conducted according to the principles of the Declaration of Helsinki. Furthermore, a requirement for informed consent was waived because this study is a retrospective analysis of existing administrative and clinical data. This manuscript was prepared following the Guidelines for Recommendations for Reporting Tumor Marker Prognostic Studies [31], as appropriate.

### 2.2. Definitions of Inflammatory and Nutritional Biomarkers

Albumin (Alb), a C-reactive protein (CRP), total cholesterol levels in serum and leucocyte, including neutrophil, lymphocyte, and monocyte, and platelet counts in peripheral blood were determined through blood analysis before the initiation of ICI therapy. Based on these results, the values of the following parameters were calculated: PLR, the ratio of platelet count to lymphocyte count; NLR, the ratio of neutrophil count to lymphocyte count; LMR, the ratio of lymphocyte count to monocyte count; SII, which is the neutrophil count × platelet count/total lymphocyte count; CAR, the ratio of the serum CRP level to serum Alb level; CONUT score, which is calculated using the serum Alb level, the total lymphocyte count, and total cholesterol level; PNI, which is 10 × serum Alb level + 0.005 × total lymphocyte count; PI, measured using the serum CRP level and leucocyte count: A CRP level > 1 mg/dL was given a score of 1, whereas a CRP level ≤ 1 mg/dL was defined as a score of 0; and GPS was measured using the serum CRP level and serum Alb level: A CRP level > 1.0 mg/dL and Alb level < 3.5 g/dL were given a score of 2, a CRP level > 1.0 mg/dL or Alb level < 3.5 g/dL was given a score of 1, and a CRP level ≤ 1.0 mg/dL and Alb level ≥ 3.5 g/dL were given a score of 0.

### 2.3. Statistical Analysis

OS and PFS were estimated using the Kaplan–Meier method and evaluated using the log-rank test. The association between ORR and each factor was assessed using the univariate and multivariate logistic regression models. The cutoff values for inflammatory and nutritional biomarkers were determined using survival Classification and Regression Tree (CART) analysis (https://cran.r-project.org/web/packages/survival/index.html (accessed on 12 Febrary 2023) and https://cran.r-project.org/web/packages/rpart/index.html (accessed on 12 Febrary 2023)) for each biomarker. The biomarkers for which results were not output by CART underwent the performance of a receiver operating characteristic (ROC) curve for OS after 12 months from the start of treatment, which identified the optimal cutoff value that had the highest sensitivity and specificity. Each biomarker was classified into two groups (high and low) by cutoff value. A Cox regression model was used to analyze the relationship between inflammatory and nutritional biomarker levels associated with OS and PFS. A multivariate analysis was performed after adjusting for age. In addition, to identify the inflammatory and nutritional biomarkers most related to OS in machine learning models, the survival CART and random survival forest (RSF) analyses [32] were conducted.

Furthermore, RSF was utilized to obtain the survival probabilities of each patient, and those with a predicted survival probability of 50% or higher at 24 months were classified as the high survival rate group, while those with less were deemed as the low survival rate group. A comparison of the overall survival rates between the two groups was conducted through a log-rank test. Finally, we computed the concordance index (c-index) for both the RSF and Cox regression model to compare the accuracy of the predictive model. Logistic regression models, cox regression model, CART, and RSF were performed using R software (ver. 4.2.2; https://www.R-project.org (accessed on 1 December 2022)). Survival probabilities and the c-index were obtained using the R-powered data tool Exploratory v6.12.2 (https::/exploratory.io (accessed on 10 March 2023)). Other statistical analyses were performed using GraphPad Prism 8 software (GraphPad Software Inc., San Diego, CA, USA). A statistically significant correlation was set at *p* < 0.05; a significant trend was defined as a *p* < 0.1.

## 3. Results

### 3.1. Patient Characteristics

From June 2017 to June 2022, 109 patients with RMHNSCC were treated with ICI treatment. Seven patients were excluded from the study because their general condition deteriorated or therapy was discontinued at the patient’s request before the first evaluation. Finally, 102 patients were enrolled in this study. The median follow-up time from ICI initiation was 13.5 months (interquartile range: 6–22). Their characteristics are summarized in Table 1. There were 93 men and nine women with a median age of 70 (47–87) years. The primary site was the hypopharynx in 40 patients, the oropharynx in 24 patients, the larynx in 12 patients, the oral cavity in 8 patients, the nasopharynx in 6 patients, and other sites in 12 patients. The ECOG PS was 0–1 in 100 patients and 2 in 2 patients. Locoregional recurrence was evident in 71 patients, distant metastases in 32 patients, and both locoregional recurrence and distant metastases in 10 patients for the evaluated disease sites. ICIs were the first line of treatment in 11 patients, the second line in 47 patients, the third line in 36 patients, and the fourth line in 8 patients. The type of ICIs was nivolumab in 76 patients and pembrolizumab in 26 patients. Prior to receiving ICI therapy, 31 patients underwent surgery, 77 patients received radiotherapy, 81 patients underwent chemotherapy, and 16 patients received cetuximab. Of the patients who received these prior treatments, 20 experienced treatment-induced immune-related adverse events, while 82 did not. There was no apparent unbalance in the patient distribution by cutoffs for each inflammatory and nutritional biomarker.

### 3.2. Treatment Outcomes

OS and PFS were estimated using the Kaplan–Meier method (Figure 1). The median OS was 21 months (95% confidence interval [CI], 17–25 months), with 1-year and 2-year OS rates of 63.6% and 42.7%, respectively (Figure 1A). Additionally, the median PFS was estimated to be seven months (95% CI, 5–9 months), with 1-year and 2-year PFS rates of 31.7% and 23.7%, respectively (Figure 1B).

The treatment response was evaluated using RECIST version 1.1. (Table 2). Among the 102 patients evaluated, a complete response (CR) was observed in 16 patients (15.7%), and a partial response (PR) was achieved in 38 patients (37.3%). A stable disease (SD) was noted in 22 patients (21.6%), and a progressive disease (PD) was observed in 26 patients (25.5%). The ORR (CR+ PR) was 52.9%, and the DCR (CR+ PR+ SD) was 74.5%.

### 3.3. Analysis of Inflammatory and Nutritional Biomarkers

The results of ORR and DCR associated with inflammation and nutritional biomarkers in ICI therapy are shown in Table 3. Regarding the determination of cutoff values, each biomarker was initially analyzed using CART, and cutoff values was output for PLR, LMR, and GPS. Cutoff values for the remaining NLR, SII, CAR, CONUT, PNI, PI, BMI, Alb, CRP, and BMI were determined using ROC curves. The cutoff values for each biomarker were as follows: age: 70 years, PLR: 397, NLR: 6.7, LMR: 1.88, SII: 107.5, CAR: 0.14, CONUT score: 3, PNI: 37.7, PI: 0, GPS: 0, body mass index (BMI): 20 kg/m^2^, Alb level: 4.1 g/dL, and CRP level: 0.89 mg/dL. No significant correlation was found between any factor and ORR. Regarding the analysis of DCR and each biomarker, significant correlations were observed between PLR, SII, and CAR in the univariate analysis. However, in the multivariate analysis, no significant differences were observed.

Table 4 shows the cox regression analysis of inflammatory and nutritional biomarkers associated with OS and PFS in patients undergoing ICI therapy. The analysis of OS and inflammatory and nutritional biomarkers showed significant correlations of OS with the Alb level, CRP level, PLR, NLR, LMR, SII, and CONUT score. In addition, multivariate analysis demonstrated that LMR was significantly correlated with OS. In the analysis of PFS and included factors, significant correlations were observed between the PFS and Alb level, CRP level, PLR, NLR, LMR, CONUT score, PNI, and GPS. Multivariate analysis showed no significant correlation; however, a significant trend in the Alb level was observed.

Figure 2 shows the results of Kaplan–Meier survival curves and the log-rank tests of cutoff values for the Alb level, PLR, NLR, LMR, and CONUT score, which were highly correlated with OS by the cox regression model. There *were significant differences in* OS *between the two groups* for each biomarker. *Among them, the LMR had the lowest p*-value *in the log-rank test.*

PLR, platelet to lymphocyte ratio; NLR, neutrophil to lymphocyte ratio; LMR, lymphocyte to monocyte ratio; CONUT, controlling nutritional status; Alb, albumin

### 3.4. Importance of Inflammatory and Nutritional Biomarkers Using Machine Learning Models

The result of the survival CART analysis is shown in Figure 3. All biomarkers were entered into CART and analyzed. As a result, LMR, CAR, BMI, Age, and PNI were selected, and the results were output as shown. The root node was first divided based on LMR, with the result that LMR was the most important factor. For a high LMR, an interaction was found between CAR, age, PNI, and OS. For a low LMR, an interaction was found for BMI.

CART, Classification and Regression Tree; LMR, lymphocyte to monocyte ratio; CAR, C-reactive protein to albumin ratio; PNI, prognostic nutrition index; BMI, body mass index

Figure 4a shows the analysis results of the variable importance (VIMP) of included biomarkers by RSF. The VIMP was obtained by measuring the decrease in prediction accuracy when randomizing a particular variable. A higher VIMP meant that the variable contributed more to predictive accuracy. The RSF demonstrated that LMR was the most important biomarker in that model, similar to the Cox regression model and CART analysis results. Moreover, PLR, NLR, and Alb were the following important biomarkers that contributed to OS in ICI-treated patients. The result of OS between the high and low survival rates group based on the prediction model by RSF showed a significant difference between the two groups in the log-rank test. By comparing the c-index by model, RSF had a c-index of 0.71, while the cox proportional hazard model had a c-index of 0.67, which was slightly higher for RSF. However, the results show that the prediction accuracy was not high for both groups.

Alb, albumin level; BMI, body mass index; CAR, CAR, CRP to albumin ratio; CONUT, controlling nutritional status; CRP, C-reactive protein; GPS, Glasgow prognostic score; LMR, lymphocyte to monocyte ratio; NLR, neutrophil to lymphocyte ratio; PI, predictive index; PLR, platelet to lymphocyte ratio; PNI, prognostic nutrition index; SII, systemic immune-inflammation index; *HSR, high survival rate group; LSR, low survival rate group; CPH, cox proportional hazard model; RSF, random survival forest*.

## 4. Discussion

With the widespread use of immunotherapy in the management of head and neck cancer, treatment outcomes and prognosis are being elucidated [5,33,34,35,36,37,38]. According to the results of this study, the ORR and DCR were 52.9% and 74.5%, respectively. Moreover, the median OS was 21 (95% CI 17-25) months, with an estimated 12-month OS rate of 63.6%. The median PFS was 7 (95% CI 5-9) months, with an estimated 12-month PFS rate of 31.7%. These results are favorable compared to the results of other published reports. While differences in the type of ICIs could be a possible explanation, the results of subgroup analysis revealed no significant differences (Appendix A). Therefore, other factors could provide a more likely reason. Another factor that was considered was the patient’s PS. As a general rule, we only included cases with a PS of 0-1, and the proportion of subjects with a PS > 2 was lower than in other reports. This could be the cause of the favorable results obtained in this study. Many reports have shown that patients with poor PS have less efficacy in ICIs [39,40,41,42]. A poor PS may indicate a poor systemic status, but biomarkers as indicators of the systemic status have not been thoroughly investigated. To date, several inflammatory and nutritional biomarkers have been reported as useful in predicting the prognosis of various carcinomas [13,15,17,20,23,27,43] and head and neck cancers after treatment with ICIs [24,30,44]. However, the prognostic value of these factors in RMHNSCC has not been thoroughly investigated. Therefore, we examined the prognostic value of these factors in RMHNSCC.

In the analysis of the correlation between ORR with ICI therapy and inflammatory and nutritional biomarkers, no significant correlation was found between any factor and ORR. However, the correlation between DCR and the included factors showed a significant correlation with SII and significant trends with PLR, LMR, PNI, PI, and CRP levels. Khaki et al. [41] reported that patients with an ECOG PS ≥ 2 had an ORR with ICI that was similar to patients with an ECOG PS of 0 to 1, but this may not compensate for the negative prognostic value of a poor PSThis result, which is consistent with our results and shows that these inflammatory and nutritional biomarkers may be related to long-term prognosis, although they are less related to ORR.

Univariate analysis for the correlation between inflammatory and nutritional biomarkers and OS showed significant correlations of the Alb level, CRP level, PLR, NLR, LMR, SII, and CONUT score with OS, while multivariate analysis showed a significant correlation only between LMR and OS. In the analysis of PFS and the included factors, univariate analysis showed significant correlations with the Alb level, CRP level, PLR, NLR, LMR, CONUT score, PNI, and GPS. In contrast, multivariate analysis revealed no significant results. Moreover, Kaplan–Meier survival curves and log-rank tests using cutoff values for the Alb level, PLR, NLR, LMR, and CONUT score demonstrated significant differences between the two groups for each biomarker. The results of this study suggest that several inflammatory and nutritional biomarkers are associated with OS and PFS in patients with RMHNSCC. Still, LMR may be the most significant prognostic factor. As an inflammatory biomarker, monocytes have been shown to regulate tumor development, progression, and metastasis [45], while tumor-associated macrophages may promote tumor cell growth, migration, and metastasis [46]. Lymphocytes are a major component of the host immune system and can eliminate cancer cells and prevent tumor progression [47]. Therefore, relatively high monocyte counts and low lymphocyte counts may predict a poor prognosis.

Several reports have been published on LMR and the prognosis of patients with RMHNSCC [21,48,49]; Kano et al. [21] reported that pretreatment with LMR could be considered an independent prognostic factor in patients with laryngeal, nasopharyngeal, and hypopharyngeal cancer undergoing chemoradiotherapy. Aoyama et al. [48] reported that the combination of LMR and PS was useful as a prognostic marker in patients with RMHNSCC treated with the EXTREME regimen. The present results indicate that LMR is also a useful biomarker in patients with RMHNSCC treated using ICIs. Other inflammatory and nutritional biomarkers, such as NLR and PLR, have also been reported as useful biomarkers for ICI therapy [20,24,30,44], but the results vary across reports. Although no precise mechanism has been elucidated, all these factors are relevant to the prognosis of RMHNSCC. The levels of inflammatory biomarkers, including LMR and PLR, and nutritional biomarkers, such as the CONUT score and GPS, which can be easily measured in blood samples, may be useful prognostic biomarkers.

We performed survival CART and random survival forest analyses [32] as machine learning models to identify the most valuable biomarkers in patients with RMHNSCC when treated using ICIs. The survival CART is a decision tree-based method used for survival analysis. The survival CART aims to build a decision tree that predicts survival time or time to an event in individuals based on a set of predictor variables. Random survival forest is a machine learning method that combines multiple decision trees in an ensemble learning process to evaluate and visualize the importance of factors. It can also predict the effectiveness of cancer treatment and survival duration. Our study results reveal that LMR was the most potent contributor to OS in both models. Together, these results suggest that LMR may be the most beneficial biomarker in patients with RMHNSCC undergoing ICI therapy. In addition, we tested the predictive model of RSF by dividing patients into two groups based on their predicted survival rates to evaluate whether the model could make such predictions. The results showed a statistically significant difference in the log-rank test between high and low survival groups. These results suggest that the model can make certain predictions, but the accuracy of this prediction was not high, with a c-index of 0.7. The small number of patients and variability were considered the main reasons for this accuracy.

This study had several limitations. First, this was a single-center retrospective study that evaluated a small number of patients. Second, the use of chemotherapy after ICI therapy was not standardized and may have been subject to bias due to differing drug selections. Third, inflammatory and nutritional biomarkers must be interpreted cautiously because of the many factors involved. Finally, a large study is needed to elucidate the prognostic biomarkers of ICI treatment for RMHNSCC in the future. As a result, the prognostic impact of biomarkers that differ across reports needs to be homogenized, and machine learning models should enable prognosis prediction.

## 5. Conclusions

We evaluated the prognostic value of inflammatory and nutritional biomarkers of ICI treatment for RMHNSCC. These biomarkers did not correlate with ORR but correlated with DCR. Univariate analysis showed significant correlations between the Alb level, CRP level, PLR, NLR, LMR, SII, CONUT score, and OS. Additionally, multivariate analysis showed that LMR was significantly correlated with OS. LMR was the most important biomarker, according to the machine learning model. This study suggests that LMR may be the most useful biomarker for predicting the prognosis of ICI treatment for RMHNSCC.

## Figures and Tables

**Figure 1 cancers-15-02021-f001:**
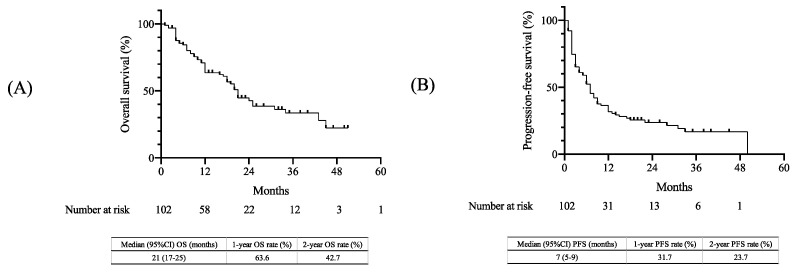
Kaplan–Meier curves in the overall population: (**A**) OS and (**B**) PFS. OS, overall survival; PFS, progression-free survival; CI, confidence interval.

**Figure 2 cancers-15-02021-f002:**
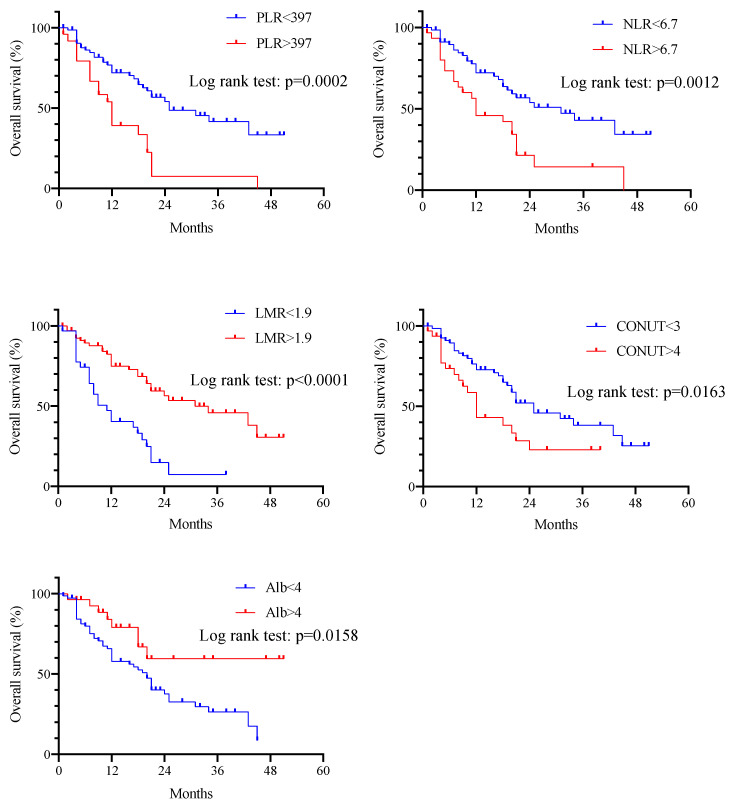
Kaplan–Meier curves for correlation between overall survival and PLR, NLR, LMR, CONUT score, and Alb level according to cutoff values. Significant differences in overall survival were observed between the two groups for each biomarker.

**Figure 3 cancers-15-02021-f003:**
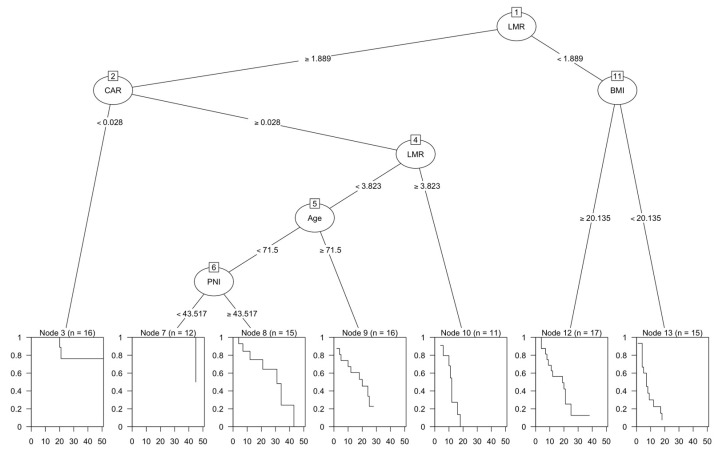
Classification and regression tree (CART) analysis. The root node was first divided based on LMR. For a high LMR, an interaction was found between CAR, age, PNI, and overall survival. For a low LMR, an interaction was found for BMI.

**Figure 4 cancers-15-02021-f004:**
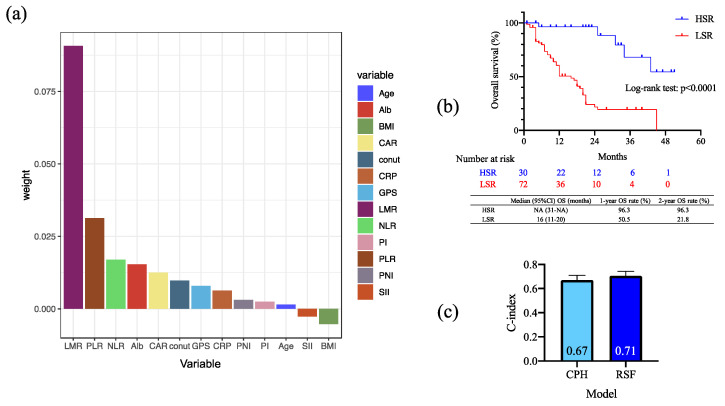
(**a**) The analysis of the importance of included factors using random survival forest (RSF). LMR had the highest weight of all factors. (**b**) Kaplan–Meier survival curves and log-rank tests by predicted survival rate using the RSF model. There was a significant difference between the two groups. (**c**) Comparison of c-index by model. RSF had a c-index of 0.71, while the cox proportional hazard model had a c-index of 0.67, which was slightly higher for RSF.

**Table 1 cancers-15-02021-t001:** Patient characteristics.

Variables	Number (n = 102)	%
Sex		
Male	93	91.2
Female	9	8.8
Median age (range)	70 (47–87)	
Primary site		
Oral	8	7.8
Nasopharynx	6	5.9
Oropharynx	24	23.5
Hypopharynx	40	39.2
Larynx	12	11.8
Others	12	11.8
ECOG performance status		
0 or 1	100	98.0
>2	2	2.0
Type of Recurrence		
Locoregional	71	69.6
Distant	21	20.6
Locoregional+Distant	10	9.8
ICI line		
1st	11	10.8
2nd	47	46.1
3rd	36	35.3
>4th	8	7.8
Previous treatment		
Nivolumab	76	74.5
Pembrolizumab	26	25.5
Prior treatment		
Surgery	31	30.4
Radiation	77	75.5
Chemo	81	79.4
Cetaximub	16	15.7
irAE		
Yes	20	19.6
No	82	80.4
PLR		
≦397	77	75.4
>397	25	24.5
NLR		
≦6.7	71	69.6
>6.7	31	30.3
LMR		
≦1.88	32	31.3
>1.88	70	68.6
SII		
≦107.5	54	52.9
>107.5	48	47
CAR		
≦0.14	52	50.9
>0.14	50	49
CONUT		
≦3	70	68.6
≧4	32	31.3
PNI		
≦37.7	18	17.6
>37.7	84	82.3
PI		
0	68	66.6
≧1	34	33.3
GPS		
0	87	85.2
≧1	15	14.7
BMI		
≦20	45	44.1
>20	57	55.8
Alb		
≦4.1	74	72.5
>4.1	28	27.4
CRP		
≦0.89	66	64.7
>0.89	36	35.2

ECOG, Eastern Cooperative Oncology Group; ICI, immune check point inhibitor; irAE, immune rerated adverse event; PLR, platelet to lymphocyte ratio; NLR, neutrophil to lymphocyte ratio; LMR, lymphocyte to monocyte ratio; SII, systemic immune-inflammation index; CAR, CRP to Alb ratio; CONUT, controlling nutritional status; PNI, prognostic nutrition index; PI, predictive index; GPS, Glasgow prognostic score; BMI, body mass index; Alb, albumin; CRP, C-reactive protein.

**Table 2 cancers-15-02021-t002:** Effectiveness of immune checkpoint inhibitor therapy.

Variables	All	%
Number of patients (%)	102	
Best response (%)		
Complete response	16	15.7
Partial response	38	37.3
Stable disease	22	21.6
Progressive disease	26	25.5
ORR	54	52.9
DCR	76	74.5

ORR, objective response rate; DCR, disease control rate.

**Table 3 cancers-15-02021-t003:** Prognostic analysis of *ORR (a) and DCR (b)* in patients who received immune checkpoint inhibitor therapy.

(a) Overall Response Rate						
		Univariate				
	Odds Ratio	95% CI	*p*-Value			
Age (<70 or >70)	0.997	0.458–2.17	0.9940			
BMI (Cutoff: 20)	0.527	0.211–1.32	0.1720			
Alb (Cutoff:4.1)	0.676	0.289–1.58	0.3650			
CRP (Cutoff:0.89)	1.35	0.582–3.11	0.4880			
PLR (Cutoff: 397)	0.736	0.337–1.61	0.4410			
NLR (Cutoff: 6.7)	0.732	0.336–1.6	0.4330			
LMR (cutoff: 1.88)	0.892	0.386–2.06	0.7890			
SII (Cutoff: 107.5)	1.44	0.518–4.02	0.4830			
CAR (Cutoff: 0.14)	0.623	0.272–1.43	0.2640			
CONUT (Cutoff: 3)	0.78	0.26–2.34	0.6570			
PNI (Cutoff: 37.7)	1.72	0.781–3.78	0.1780			
PI (Cutoff: 0)	2.01	0.817–4.92	0.1290			
GPS (Cutoff: 0)	0.627	0.277–1.42	0.2630			
**(b) Disease control rate**						
		Univariate			Multivariate	
	Odds ratio	95% CI	*p*-value	Odds ratio	95% CI	*p*-value
Age (<70 or >70)	0.452	0.179–1.14	0.0925	0.391	0.145–1.06	0.651
BMI (Cutoff: 20)	0.393	0.149–1.04	0.0598	1.82	0.682–4.86	0.232
Alb (Cutoff:4.1)	0.487	0.192–1.24	0.1300			
CRP (Cutoff:0.89)	2.4	0.952–6.05	0.0635			
PLR (Cutoff: 397)	0.365	0.144–0.923	**0.0332**	0.397	0.117–1.41	0.157
NLR (Cutoff: 6.7)	0.626	0.255–1.54	0.3070			
LMR (cutoff: 1.88)	0.521	0.206–1.31	0.1670			
SII (Cutoff: 107.5)	2.93	1.01–8.52	**0.0478**	0.613	0.186–2.02	0.420
CAR (Cutoff: 0.14)	0.382	0.152–0.957	**0.0399**	0.619	0221–1.74	0.360
CONUT (Cutoff: 3)	0.448	0.142–1.41	0.1700			
PNI (Cutoff: 37.7)	2.09	0.847–5.16	0.1100			
PI (Cutoff: 0)	2.54	0.788–8.18	0.1190			
GPS (Cutoff: 0)	0.434	0.174–1.08	0.0726			

ORR, objective response rate; DCR, disease response rate; PLR, platelet to lymphocyte ratio; NLR, neutrophil to lymphocyte ratio; LMR, lymphocyte to monocyte ratio; SII, systemic immune-inflammation index; CAR, CRP to Alb ratio; CONUT, controlling nutritional status; PNI, prognostic nutrition index; PI, predictive index; GPS, Glasgow prognostic score; BMI, body mass index; Alb, albumin; CRP, C-reactive protein. Bold indicates statistically significant *p*-values.

**Table 4 cancers-15-02021-t004:** Cox regression analysis for correlation of nutritional and inflammatory biomarkers associated with overall survival and progression-free survival.

Overall Survival						
		Univariate			Multivariate	
	Hazard Ratio	95% CI	*p*-Value	Hazard Ratio	95% CI	*p*-Value
Age (<70 or >70)	0.7737	0.4308–1.389	0.3904	1.189	0.6783–2.086	0.545
BMI (Cutoff: 20)	0.8211	0.4776–1.412	0.4757			
Alb (Cutoff:4.1)	0.4133	0.1938–0.8813	**0.0222**	0.5297	0.2353–1.192	0.1248
CRP (Cutoff:0.89)	1.792	1.039–3.093	**0.0361**			
PLR (Cutoff: 397)	2.77	1.568–4.891	**0.0004**	1.776	0.7779–4.057	0.1726
NLR (Cutoff: 6.7)	2.349	1.361–4.053	**0.0022**	0.9184	0.4027–2.095	0.8397
LMR (cutoff: 1.88)	0.3117	0.1776–0.5473	**0.0000**	0.4275	0.2073–0.8816	**0.02139**
SII (Cutoff: 107.5)	1.774	1.023–3.079	**0.0414**			
CAR (Cutoff: 0.14)	1.479	0.8541–2.56	0.1625			
CONUT (Cutoff: 3)	1.942	1.104–3.417	**0.0213**	1.083	0.5665–2.069	0.8098
PNI (Cutoff: 37.7)	0.543	0.2887–1.022	0.0582			
PI (Cutoff: 0)	1.729	0.9952–3.005	0.0520			
GPS (Cutoff: 0)	1.653	0.847–3.227	0.1407			
**Progression-Free Survival**						
		Univariate			Multivariate	
	Hazard ratio	95% CI	*p*-value	Hazard ratio	95% CI	*p*-value
Age (<70 or >70)	0.9701	0.6128–1.536	0.8969	0.9409	0.5804–1.525	0.8047
BMI (Cutoff: 20)	0.9565	0.6021–1.519	0.8504			
Alb (Cutoff 4.1)	0.4661	0.2636–0.8244	**0.0087**	0.5468	0.2933–1.019	0.05749
CRP (Cutoff 0.89)	1.644	1.019–2.653	**0.0416**			
PLR (Cutoff: 397)	1.899	1.152–3.133	**0.0120**	1.396	0.6435–3.028	0.3986
NLR (Cutoff: 6.7)	1.773	1.103–2.85	**0.0180**	1.156	0.5495–2.434	0.7019
LMR (cutoff: 1.88)	0.6033	0.372–0.9786	**0.0406**			
SII (Cutoff: 107.5)	1.364	0.861–2.162	0.1859			
CAR (Cutoff: 0.14)	1.483	0.9355–2.352	0.0937			
CONUT (Cutoff: 3)	1.848	1.131–3.019	**0.0142**	0.8504	0.4017–1.8	0.672
PNI (Cutoff: 37.7)	0.3622	0.2048–0.6404	**0.0005**	0.3856	0.07537–1.973	0.2525
PI (Cutoff: 0)	1.492	0.9179–2.425	0.1065			
GPS (Cutoff: 0)	2.602	1.437–4.709	**0.0016**	0.8807	0.1881–4.123	0.8718

PLR, platelet to lymphocyte ratio; NLR, neutrophil to lymphocyte ratio; LMR, lymphocyte to monocyte ratio; SII, systemic immune-inflammation index; CAR, CRP to Alb ratio; CONUT, controlling nutritional status; PNI, prognostic nutrition index; PI, predictive index; GPS, Glasgow prognostic score; BMI, body mass index; Alb, albumin; CRP, C-reactive protein. Bold indicates statistically significant *p*-values.

## Data Availability

All relevant data have been presented in this manuscript.

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
