# Peer review of "Prognostic Value of Inflammatory and Nutritional Biomarkers of Immune Checkpoint Inhibitor Treatment for Recurrent or Metastatic Squamous Cell Carcinoma of the Head and Neck"

_cancers, 2023, doi:10.3390/cancers15072021_

Round 1

Reviewer 1 Report

In this manuscript, the authors provide an interesting retrospective analysis of patients having recurrent or metastatic squamous cell carcinoma of the head and neck (RMHNSCC) and treated with immune checkpoint inhibitor (ICI). Lymphocyte-to-monocyte ratio (LMR) is identified as a possible biomarker predicting prognosis of an ICI therapy for RMHNSCC. Yet, for the reader’s ease of understanding, the following additions are strongly recommended before publication.

In or before Section 3.4, it is recommended to detail how “[t]he cutoff values for inflammatory and nutritional biomarkers were determined using receiver operating characteristic curves or survival Classification and Regression Tree (CART) analysis.” In another word, shall Section 3.5 be presented before Section 3.4? Why were LMR, CAR, BMI, age and PNI selected for the CART analysis? Why were Alb, CONUT, NLR and PLR (which are the focuses of Figures 2 and 4) excluded from the CART analysis or not shown in the final tree of Figure 3?

Further, kindly specify (preferably in the introduction section) whether the ICI as discussed is limited to PD-1/PD-L1 inhibitors, or more specifically, Nivolumab and Pembrolizumab. In addition, the sentence starting in line 277 said “[s]everal reports have been published on LMR and the prognosis of patients with RMHNSCC.” Does this statement align with the purpose of this study as disclosed in the introduction section? How does the current manuscript advance our knowledge about biomarkers and RMHNSCC then? Also, further to the discussion section, I am curious why ECOG PS was not included in the analysis.

Another general (and maybe hypercritical) comment I have here is about Sections 3.1, 3.2, and 3.3. The result description there seems a line-by-line repeat of all the information presented in corresponding tables. Just wondering, is it absolutely needed?

In addition, the following minor changes would be really appreciated.

(1)         Full names of acronyms need to be provided on their first appearances, such as RMHNSCC and LMR.

(2)         A brief introduction of the Checkmate 141 and Keynote 048 studies is suggested in the introduction section.

(3)         There are three “and” after the term “between” in lines 178-179 as well as in lines 187-188, which may cause ambiguousness. Similar comment to the uses of “and” across the manuscript.

(4)         kindly elaborate on how “[t]he most appropriate ICI was selected at our multidisciplinary head and neck cancer conference based on each patient's condition” as stated in lines 66-67.

(5)         Kindly clarify the differences between “significant correlation” and “significant trends” as stated in lines 167-168, 185, and 253-254.  

Author Response

Point-by-point Response to Reviewer's Comments 
The revised manuscript has improved its scientific integrity and responded to questions/comments provided during the interactive discussion. We wish to express our appreciation to the Reviewers for their insightful comments, which have helped us significantly improve the paper.

We have responded to the point-by-point responses below and highlighted changes in the manuscript.

Reviewer Comments:
Reviewer: 1
In this manuscript, the authors provide an interesting retrospective analysis of patients having recurrent or metastatic squamous cell carcinoma of the head and neck (RMHNSCC) and treated with immune checkpoint inhibitor (ICI). Lymphocyte-to-monocyte ratio (LMR) is identified as a possible biomarker predicting prognosis of an ICI therapy for RMHNSCC. Yet, for the reader's ease of understanding, the following additions are strongly recommended before publication.
In or before Section 3.4, it is recommended to detail how "the cutoff values for inflammatory and nutritional biomarkers were determined using receiver operating characteristic curves or survival Classification and Regression Tree (CART) analysis." In another word, shall Section 3.5 be presented before Section 3.4? Why were LMR, CAR, BMI, age and PNI selected for the CART analysis? Why were Alb, CONUT, NLR and PLR (which are the focuses of Figures 2 and 4) excluded from the CART analysis or not shown in the final tree of Figure 3?
Response: Thank you for your valuable feedback and suggestions. We apologize for any confusion caused by the Classification and Regression Tree (CART) analysis. The first step of the CART analysis involves identifying the most effective explanatory variable for categorizing the outcome and then segmenting the patients based on this variable. The resulting subgroups are further partitioned using the same approach until no further partitioning results in enhanced classification accuracy or a pre-defined stopping criterion is satisfied. The final subtree is then pruned to minimize the rate of misclassification. In other words, the CART analysis selects the best cutoff value for a single factor but will not output the result if the factor cannot be split well. Thus, the Receiver Operating Characteristic (ROC) curve is used to select the cutoff value for such factors, and we have revised the method section in the main text regarding the selection of cutoff values.
When multiple factors are analyzed, the factors are automatically selected and output. In this study, all factors were entered into CART and analyzed. As a result, LMR, CAR, BMI, Age, and PNI were selected, and the results were output, as shown in figure 3.
Furthermore, the study's main content is up to 3.4,. In addition, we were interested in what results the machine learning model would produce, so we performed a survival CART analysis and a random survival forest analysis. 
The findings indicated that, although the critical factors varied for each analysis, LMR was the most crucial predictor in both tests. We incorporated these results into the main text as section 3.5, which was supported by the primary findings. Nevertheless, we are open to changing the order and will gladly change it to 3.5 before 3.4 if you prefer. 

Revised Manuscript (Line 217-223):
2.3. Statistical analysis
The cutoff values for inflammatory and nutritional biomarkers were determined using survival Classification and Regression Tree (CART) analysis (https://cran.r-project.org/web/packages/survival/index.html and https://cran.r-project.org/web/packages/rpart/index.html) for each biomarker. For biomarkers for which results were not output by CART, the receiver operating characteristic curve for OS after 12 months from the start of treatment was performed to identify the optimal cutoff value with the highest sensitivity and specificity.
…….
Revised Manuscript (Line 473-476):
Figure 3 shows the result of the survival CART analysis. All biomarkers were entered into CART and analyzed. As a result, LMR, CAR, BMI, Age, and PNI were selected, and the results were output as shown. The root node was first divided based on LMR. For a high LMR, an interaction was found between CAR, age, PNI, and OS. For a low LMR, an interaction was found for BMI.

Further, kindly specify (preferably in the introduction section) whether the ICI, as discussed is limited to PD-1/PD-L1 inhibitors, or more specifically, Nivolumab and Pembrolizumab. 
Response: Thank you for your insightful comments and questions. Reviewer 2 also suggested that we should include a description of the ICI in the introduction. The introduction has been changed and added.

Revised Manuscript (Line 45-54):
1.    Introduction

RMHNSCC because they are associated with fewer adverse events [4,5] and better patient quality of life than chemotherapy. NCCN Guidelines for Head and Neck Cancers V.1.2023 recommend two ICIs, nivolumab, and pembrolizumab, as preferred regimens. CheckMate 141 study [2] was a clinical trial that evaluated the effectiveness of nivolumab for RMHNSCC. The trial compared nivolumab to chemotherapy and found that patients who received nivolumab had longer overall survival rates and fewer adverse events than chemotherapy. Also, the Keynote-048 trial [3] reported the effectiveness of the pembrolizumab as a first-line treatment for RMHNSCC. Pembrolizumab alone and pembrolizumab plus chemotherapy demonstrated superior OS and progression-free survival (PFS) compared to chemotherapy, leading to the approval of pembrolizumab as a first-line treatment for RMHNSCC. However, the response rates in the Checkmate 141 study [1] and the Keynote 048 study [2] were 13.3% and 16.9%, respectively, and not many patients benefit from ICIs. Therefore, es-timating the effects of ICIs and the prognosis after ICI treatment are crucial to determine the optimal treatment strategy.

In addition, the sentence starting in line 277 said "several reports have been published on LMR and the prognosis of patients with RMHNSCC." Does this statement align with the purpose of this study as disclosed in the introduction section? How does the current manuscript advance our knowledge about biomarkers and RMHNSCC then? 
Response: Thank you for your insightful comments. There have been several reports that LMR is associated with prognosis in head and neck cancer patients and after chemotherapy in RMHNSCC. However, there is no report to date that LMR is a biomarker for ICI treatment in RMHNSCC. Therefore, we believe that the finding that LMR is currently a useful prognostic biomarker for ICI treatment is very important. For clarity, the introduction has been revised.

Revised Manuscript (Line 58-73):
1.    Introduction
…….
In recent years, various biomarkers have been developed to assist in the selection of anticancer agents [6,7], including PD-L1 expression [8,9], tumor mutation burden [10], interferon-γ signature [11], and tumor microenvironment [12]. However, most of these biomarkers are research-based and not suitable for clinical application. In contrast, several inflammatory and nutritional biomarkers have been reported to be associated with prognosis [13-17]. In particular, neutrophil-to-lymphocyte ratio (NLR) is a well-known biomarker and has been reported to be an independent prognostic factor in head and neck cancer [18]. In addition, many inflammatory and nutritional biomarkers that can be easily measured from blood tests, such as platelet to lymphocyte ratio (PLR) [19,20], lymphocyte to monocyte ratio (LMR) [21], systemic immune-inflammation index (SII) [22,23], CRP to Alb ratio (CAR) [24] [25], controlling nutritional status (CONUT) score[26] [14], prognostic nutrition index (PNI) [27] [28], prognostic index (PI) [29], and the Glasgow Prognostic Score (GPS) [13], have been reported as useful for predicting prognosis after treatment in head and neck and other cancers [19,20,24,25,30]. However, it is not clear which biomarkers best predict prognosis after ICI treatment and their prognostic value in RMHNSCC has not been fully investigated.
Therefore, the purpose of this study was to determine the prognostic value of inflammatory and nutritional biomarkers, the levels of which can be measured by blood analysis, in the treatment of RMHNSCC with ICIs and to identify the most useful factor for prognosis assessment.

Also, further to the discussion section, I am curious why ECOG PS was not included in the analysis.

Response: Thank you for your insightful comments and questions. We appreciate your curiosity about the potential association of ECOG PS with the outcome of interest. However, we did not analyze PS in this study. The reason being, a majority of our patient background consisted of PS 0 or 1, with only 2% having PS of 2 or higher, generally classified as poor. Hence, PS was not examined in this study. Nonetheless, we do acknowledge ECOG PS as an important prognostic factor in cancer treatment and have discussed its significance by comparing it with other studies in the discussion.

Another general (and maybe hypercritical) comment I have here is about Sections 3.1, 3.2, and 3.3. The result description there seems a line-by-line repeat of all the information presented in corresponding tables. Just wondering, is it absolutely needed?
Response: Thank you for your feedback and comments. We appreciate your attention to detail and agree that some of the information in Sections 3.1, 3.2, and 3.3 may be repetitive with the corresponding tables. As you suggested, sections 3.1-3.3 have been revised.

Revised Manuscript (Line 289-308 and 328-346):
3.1. Patient characterisics
From June 2017 to June 2022, 109 patients with RMHNSCC were treated with ICIs treatment. Among them, seven patients were excluded from the study because their general condition deteriorated or treatment was discontinued at the patient's request before the first evaluation. Finally, 102 patients were enrolled in this study. Median follow-up time from ICI initiation was 13.5 months (interquartile range: 6–22). Their characteristics are summarized in Table 1. There were 93 men and nine women with a median age of 70 (47-87) years. The primary site was the hypopharynx in 40 patients, oropharynx in 24 patients, larynx in 12 patients, oral cavity in 8 patients, nasopharynx in 6 patients, and other than these sites in 12 patients. The ECOG PS was 0-1 in 100 patients and 2 in 2 patients. Locoregional recurrence was evident in 71 patients, distant metastases in 32 patients, and both locoregional recurrence and distant metastases in 10 patients for the evaluated disease sites. ICIs were the first line of treatment in 11 patients, second line in 47 patients, third line in 36 patients, and fourth line in 8 patients. The type of ICIs was nivolumab in 76 patients and pembrolizumab in 26 patients. Prior to receiving ICI therapy, 31 patients underwent surgery, 77 patients received radiotherapy, 81 patients underwent chemotherapy, and 16 patients received cetuximab. Of the patients who received these prior treatments, 20 experienced treatment-induced immune-related adverse events, while 82 did not. There was no apparent unbalance in the patient distribution by cutoffs for each inflammatory and nutritional biomarkers. 
3.2. Treatment outcomes
OS and PFS were estimated using the Kaplan-Meier method (Figure 1). The median OS was 21 months (95% confidence interval [CI], 17-25 months), with 1-year and 2-year OS rates of 63.6% and 42.7%, respectively (Figure 1A). Additionally, the median PFS was estimated to be 7 months (95% CI, 5-9 months), with 1-year and 2-year PFS rates of 31.7% and 23.7%, respectively (Figure 1B).
The treatment response was evaluated using RECIST version 1.1. (Table 2). Among the 102 patients evaluated, a complete response (CR) was observed in 16 patients, (15.7%) and a partial response (PR) was achieved in 38 patients (37.3%). A stable disease (SD) was noted in 22 patients (21.6%) and a progressive disease (PD) was observed in 26 patients (25.5%). The ORR (CR+ PR) was 52.9%, and the DCR (CR+ PR+ SD) was 74.5%.

In addition, the following minor changes would be really appreciated.
(1)    Full names of acronyms need to be provided on their first appearances, such as RMHNSCC and LMR.
Response: We have added the full names of acronyms RMHNSCC and LMR on their first appearances.

Revised Manuscript (Line 38-39 and 66-67):
Head and neck squamous cell carcinomas constitute the sixth most common cancer type in the world [1]. Despite improvements of the treatment, recurrence and metastasis are common and contribute to poor prognosis. Therefore, improving the prognosis of recurrent or metastatic squamous cell carcinoma of the head and neck (RMHNSCC) patients is crucial for the treatment of RMHNSCC
.…….
In addition, many inflammatory and nutritional biomarkers that can be easily measured from blood tests, such as platelet to lymphocyte ratio (PLR) [19,20], lymphocyte to monocyte ratio (LMR) [21], systemic immune-inflammation index (SII) [22,23], CRP to Alb ratio (CAR) [24] [25], controlling nutritional status (CONUT) score[26] [14], prognostic nutrition index (PNI) [27] [28], prognostic index (PI) [29], and the Glasgow Prognostic Score (GPS) [13], have been reported as useful for predicting prognosis after treatment in head and neck and other cancers [19,20,24,25,30].

(2)         A brief introduction of the Checkmate 141 and Keynote 048 studies is suggested in the introduction section.
Response: We have included a brief explanation of the Checkmate 141 and Keynote 048 studies in the introduction section as above.

Revised Manuscript (Line 45-54):

RMHNSCC because they are associated with fewer adverse events [4,5] and better patient quality of life than chemotherapy. Currently, NCCN Guidelines for Head and Neck Cancers V.1.2023 recommend two ICIs, nivolumab and pembrolizumab, as preferred regimens. CheckMate 141 study [2] was a clinical trial that evaluated the effectiveness of nivolumab for RMHNSCC. The trial compared nivolumab to chemotherapy and found that patients who received nivolumab had longer overall survival rates and fewer ad-verse events than chemotherapy. Also, the Keynote-048 trial [3] reported the effectiveness of the pembrolizumab as a first-line treatment for RMHNSCC. Pembrolizumab alone and pembrolizumab plus chemotherapy demonstrated superior OS and progression-free survival (PFS) compared to chemotherapy, leading to the approval of pembrolizumab as a first-line treatment for RMHNSCC. However, the response rates in the Checkmate 141 study [2]and the Keynote 048 study [3] were 13.3% and 16.9%, respectively, and not many patients benefit from ICIs. Therefore, estimating the effects of ICIs and the prognosis after ICI treatment are crucial to determine the optimal treatment strategy.

(3)         There are three "and" after the term "between" in lines 178-179 as well as in lines 187-188, which may cause ambiguousness. Similar comment to the uses of "and" across the manuscript.
Response: We have revised the sentences to avoid ambiguity and reduce the use of the term "and." In addition, we have revised minor corrections throughout the manuscript.

Revised Manuscript (Line 425-426 and 444-445):
Table 4 shows the cox regression analysis of inflammatory and nutritional biomarkers associated with OS and PFS in patients who had undergone ICI therapy. 

Table 4. Cox regression analysis for correlation of nutritional and inflammatory biomarkers associated with overall survival and progression free survival
et. al…

(4)         kindly elaborate on how "the most appropriate ICI was selected at our multidisciplinary head and neck cancer conference based on each patient's condition" as stated in lines 66-67.
Response: We have added more information on how the most appropriate ICI was selected at our multidisciplinary head and neck cancer conference.

Revised Manuscript (Line 88-90):
The most appropriate ICI was selected at our multidisciplinary head and neck cancer conference based on each patient's condition. In principle, nivolumab was indicated for treating platinum-refractory RMHNSCC and pembrolizumab was indicated for the first-line treatment of RMHNSCC or platinum-sensitive RMHNSCC. 

(5)         Kindly clarify the differences between "significant correlation" and "significant trends" as stated in lines 167-168, 185, and 253-254.  
Response: We have clarified the differences between "significant correlation" and "significant trends" in the manuscript

Revised Manuscript (Line 285-286):
Statistical significance correlation was set at p < 0.05 and significance trend was defined as a P <0.1.

Sincerely,

Akihiro Sakai

Reviewer 2 Report

1. Introduction can be slightly extended. Consider integrate information from these articles: 10.1111/jop.13264 and little bit more should be introduced about RMHNSCC and ICI. You could see an example here: https://www.ncbi.nlm.nih.gov/pmc/articles/PMC8269333/ how to add general concept, introduce problems and propose your research topic.

2. Please, integrate REMARK guidelines and do the changes based on checklist. 

3. Change PNI to a different short name since it is commonly used as perineural invasion

4. Check NLR based on these considerations: 10.1111/jop.13264

5. I believe a uni/multivariate logistic regression could be performed to investigate variables to predict therapy response outcome.

6. HPV should be investigated and evaluated in the analyses, since HPV tumors have notable better prognosis and treatment outcomes. 

7. Remove abreviations from simple summary 

8. Authors should be able to better show results from random survival ML, by dividing patients in high and low risk based on adopted model. The C-index performance could be evaluated and compared to other models.

9. Ethical approval?

Discussion is well done

The Manuscript is of strong interest, authors have been able to collect data which are difficult to retrieve in most scenarios and analyzed with proper methods. Only small changes are required and consirations

Author Response

Point-by-point Response to Reviewer's Comments 
The revised Manuscript has improved its scientific integrity and responded to questions/comments provided during the interactive discussion. We wish to express our appreciation to the Reviewers for their insightful comments, which have helped us significantly improve the paper.

We have responded to the point-by-point responses below and highlighted changes in the Manuscript.

Reviewer Comments:
Reviewer: 2
Comments and Suggestions for Authors
1.    Introduction can be slightly extended. Consider integrate information from these articles: 10.1111/jop.13264 and little bit more should be introduced about RMHNSCC and ICI. You could see an example here: https://www.ncbi.nlm.nih.gov/pmc/articles/PMC8269333/ how to add general concept, introduce problems and propose your research topic.
Response: Thank you for your insightful suggestions. Reviewer 1 also suggested that we should include a description of the ICI in the introduction. The introduction has been changed and added as you suggested.

Revised Manuscript (Line 35-56 and 60-73):
1. Introduction
Head and neck squamous cell carcinomas constitute the sixth most common cancer type in the world [1]. Despite improvements of the treatment, recurrence and metastasis are common and contribute to poor prognosis. Therefore, improving the prognosis of recurrent or metastatic squamous cell carcinoma of the head and neck (RMHNSCC) patients is crucial for the treatment of RMHNSCC. Recently approved immune checkpoint inhibitors (ICIs) have significantly improved the prognosis of patients with recurrent or metastatic squamous cell carcinoma of the head and neck (RMHNSCC) compared to conventional chemotherapy [2,3]. ICIs have been widely used as the first-line treatment for RMHNSCC because they are associated with fewer adverse events [4,5] and better patient quality of life than chemotherapy. Currently, NCCN Guidelines for Head and Neck Cancers V.1.2023 recommend two ICIs, nivolumab and pembrolizumab, as preferred regimens. CheckMate 141 study [2] was a clinical trial that evaluated the effectiveness of nivolumab for RMHNSCC. The trial compared nivolumab to chemotherapy and found that patients who received nivolumab had longer overall survival rates and fewer adverse events than chemotherapy. Also, the Keynote-048 trial [3] reported the effectiveness of the pembrolizumab as a first-line treatment for RMHNSCC. Pembrolizumab alone and pembrolizumab plus chemotherapy demonstrated superior OS and progression-free survival (PFS) compared to chemotherapy, leading to the approval of pembrolizumab as a first-line treatment for RMHNSCC. However, the response rates in the Checkmate 141 study [2]and the Keynote 048 study [3] were 13.3% and 16.9%, respectively, and not many patients benefit from ICIs. Therefore, estimating the effects of ICIs and the prognosis after ICI treatment are crucial to determine the optimal treatment strategy.
………
However, most of these biomarkers are research-based and not suitable for clinical ap-plication. In contrast, several inflammatory and nutritional biomarkers have been re-ported to be associated with prognosis [13-17]. In particular, neutrophil-to-lymphocyte ratio (NLR) is a well-known biomarker and has been reported to be an independent prognostic factor in head and neck cancer [18]. In addition, many inflammatory and nutritional biomarkers that can be easily measured from blood tests, such as platelet to lymphocyte ratio (PLR) [19,20], lymphocyte to monocyte ratio (LMR) [21], systemic immune-inflammation index (SII) [22,23], CRP to Alb ratio (CAR) [24] [25], controlling nutritional status (CONUT) score[26] [14], prognostic nutrition index (PNI) [27] [28], prognostic index (PI) [29], and the Glasgow Prognostic Score (GPS) [13], have been re-ported as useful for predicting prognosis after treatment in head and neck and other cancers [19,20,24,25,30]. However, it is not clear which biomarkers best predict prognosis after ICI treatment and their prognostic value in RMHNSCC has not been fully investigated.

2.    Please, integrate REMARK guidelines and do the changes based on checklist. 
Response: Thank you for your suggestion. As you indicated, we have added and revised the Manuscript, figures and tables as much as possible according to the REMARK guidelines and checklist. Please refer the highlighting changes in the Manuscript.

Revised Manuscript (Line 109-115):
1. Introduction

3. Change PNI to a different short name since it is commonly used as perineural invasion
Response: Thank you for your suggestion. I understand PNI is commonly used as perineural invasion. However, previous report abbreviated the prognostic nutrition index as PNI (doi: 10.3389/fcell.2021.656741, https://doi.org/10.2147/JIR.S338421, https://link.springer.com/article/10.1007/s00595-017-1582-y , et. al.), so we used PNI because we thought it would be confusing to use a new and different abbreviation. 

4. Check NLR based on these considerations: 10.1111/jop.13264

Response: The introduction has been changed and a reference, as you suggested, has been added.

5. I believe a uni/multivariate logistic regression could be performed to investigate variables to predict therapy response outcome.
Response: Thank you for your insightful comments. As our analysis method was mistaken, we have corrected it and changed Table 3 by using uni/multivariate logistic regression for the analysis on response rate according to your suggestions. We would be grateful if you could verify the highlighted sections. Thank you. 

Revised Manuscript (Line 214-223):
Table 3 Prognostic analysis of ORR (a) and DCR (b) in patients who received immune checkpoint inhibitor therapy

6. HPV should be investigated and evaluated in the analyses, since HPV tumors have notable better prognosis and treatment outcomes. 
Response: Thank you for your thoughtful review and valuable feedback. I understand that HPV status has a significant impact on the prognosis of patients with head and neck cancer. However, the impact of ICI treatment on prognosis is not as significant. As reported by Cohen et al. "Overall, HPV status should not affect the selection of patients with platinum-refractory R/M HNSCC for ICI therapy. The subcommittee noted a lack of strong data suggesting p16+ patients experience a distinct benefit and that data thus far indicate that both p16+ and p16- populations benefit from available checkpoint inhibitors." (https://jitc.bmj.com/content/7/1/184).

That being said, we present the results regarding the impact of HPV status on prognosis, as we understand there is an interest in this area. Although we observed some differences, we did not identify any significant advantages. We have prepared a separate paper on the clinical factors, including HPV and prognosis, and intend to reveal our results there. This paper solely presents the analysis results using factors that can be easily measured by blood tests. However, if the results from this study are deemed necessary for this paper, we will incorporate them as supplementary data.

7. Remove abbreviations from simple summary 
Response: We have removed abbreviations from simple summary (Line 16)

8. Authors should be able to better show results from random survival ML, by dividing patients in high and low risk based on adopted model. The C-index performance could be evaluated and compared to other models.

Response: Thank you for your insightful suggestion. As you indicated, we have added the analysis regarding RSF and revised Figure 4. Please refer the highlighting changes regarding Figure 4 in the Manuscript.

Revised figure 4:

Figure 4. (a) The analysis of the importance of included factors using random survival forest (RSF). LMR had the highest weight of all factors. (b) Kaplan-Meier survival curves and log-rank tests by predicted survival rate using RSF model. There was a significant difference between the two groups. (c) Comparison of c-index by model. RSF had a c-index of 0.71, while cox proportional hazard model had a c-index of 0.67, which was slightly higher for RSF.

9. Ethical approval?
Response: Yes. This study was approved by the institutional review board of our university.

Revised Manuscript (Line 190-191):
The Institutional Review Board of Tokai University Hospital (22R200) approved this study, which was conducted according to the principles of the Declaration of Helsinki.
Discussion is well done
The Manuscript is of strong interest, authors have been able to collect data which are difficult to retrieve in most scenarios and analyzed with proper methods. Only small changes are required and consirations

Sincerely,

Akihiro Sakai

Round 2

Reviewer 1 Report

Many thanks for the authors' amendments. Really appreciate them. I only have some very minor suggestions left. See, the details below.

(1) Among all cutoff values, which were output by CART and which were determined using ROC curves?

(2) The "OS" in line 52 is believed to be the first appearance of this acronym thus its full name is needed to be provided there instead of in line 108.

(3) The phrase "the relationship between inflammatory and nutritional biomarker levels and OS and PFS" spanning lines 155-156 is suggested to be polished.

(4) The phrase "the significant trend" in line 173 is suggested to be changed to "a significant trend."

(5) The current figure 3 is very blurry.

Author Response

Reviewer Comments:
Reviewer: 1

Comments and Suggestions for Authors
Many thanks for the authors' amendments. Really appreciate them. I only have some very minor suggestions left. See, the details below.
Thank you for taking the time to review our manuscript and providing such insightful and constructive feedback. We appreciate your kind words about the improvements we have made to the paper. We have responded to the point-by-point responses below and highlighted changes in the manuscript.

(1)    Among all cutoff values, which were output by CART and which were determined using ROC curves?

Response: Thank you for your comment. We apologize for the inadequacy of the cutoff value settings you pointed out in the last issue. We have added the part you pointed out to the manuscript. We appreciate your feedback.

Revised Manuscript (Page 8, Line 534-537):
3.3. Analysis of inflammatory and nutritional biomarkers
The results of the association between ORR and DCR and inflammatory and nutritional biomarkers in ICI therapy are shown in Table 3. Regarding the determination of cutoff values, each biomarker was initially analyzed using CART, and cutoff values were output for PLR, LMR, and GPS. Cutoff values for the remaining NLR, SII, CAR, CONUT, PNI, PI, BMI, Alb, CRP, and BMI were determined using ROC curves. The cutoff values for each biomarker were as follows: age: 70 years, PLR: 397, NLR: 6.7, LMR: 1.88, SII: 107.5, CAR: 0.14, CONUT score: 3, PNI: 37.7, PI: 0, GPS: 0, body mass index (BMI): 20 kg/m2, Alb level: 4.1 g/dL, and CRP level: 0.89 mg/dL. No significant correlation was found between any factor and ORR. 

(2)    The "OS" in line 52 is believed to be the first appearance of this acronym thus its full name is needed to be provided there instead of in line 108.
Response: Thank you for pointing this out. We will ensure to provide the full name of "OS" at its first appearance in the manuscript.

Revised Manuscript (Page 2, Line 59):
1. Introduction
Pembrolizumab alone and pembrolizumab plus chemotherapy demonstrated superior overall survival (OS) and progression-free survival (PFS) compared to chemotherapy, leading to the approval of pembrolizumab as a first-line treatment for RMHNSCC.

Revised Manuscript (Page 3, Line 215): 
2.1. Patient and data collection
The disease control rate (DCR) was defined as the percentage of patients with CR, PR, or stable disease (SD) as the best response. OS was defined as the time from the start of treatment to the date of death, regardless of the cause, or cutoff. 

(3)    The phrase "the relationship between inflammatory and nutritional biomarker levels and OS and PFS" spanning lines 155-156 is suggested to be polished.

Response: Thank you very much for your suggestion. We have changed the sentence wording.

Revised Manuscript (Page 3, Line 257-259):
A Cox regression model was used to analyze the relationship of inflammatory and nutritional biomarker levels associated with OS and PFS.

(4)    The phrase "the significant trend" in line 173 is suggested to be changed to "a significant trend."

Response: Thank you for your suggestion. We will make the necessary changes and replace "the significant trend" with "a significant trend."

(5) The current figure 3 is very blurry.

Response: We apologize for the poor quality of Figure 3. We will ensure that the figure is of high quality and resolution in the final version of the manuscript.

Sincerely,

Akihiro Sakai
